# Evaluation of the Nutritional Status of Gaucher Disease Type I Patients under Enzyme Replacement Treatment

**DOI:** 10.3390/nu14153180

**Published:** 2022-08-03

**Authors:** Paola Iaccarino Idelson, Enza Speranza, Maurizio Marra, Fabrizio Pasanisi, Rosa Sammarco, Ferruccio Galletti, Pasquale Strazzullo, Antonio Barbato

**Affiliations:** Department of Clinical Medicine and Surgery, Federico II University of Naples Medical School, 80131 Naples, Italy; enza.speranza@unina.it (E.S.); maurizio.marra@unina.it (M.M.); pasanisi@unina.it (F.P.); rosa.sammarco@unina.it (R.S.); galletti@unina.it (F.G.); strazzul@unina.it (P.S.); abarbato@unina.it (A.B.)

**Keywords:** Gaucher disease, nutritional status, energy metabolism, resting energy expenditure, indirect calorimetry

## Abstract

(1) Background: Gaucher disease (GD) is a rare lysosomal storage disease. The few studies analyzing Resting Energy Expenditure (REE) in GD involved mainly untreated patients and supported a hypermetabolic condition possibly due to the associated inflammatory state. Definitive conclusions could not be drawn also because of the heterogeneity and the small size of the samples investigated. In order to expand current knowledge concerning, in particular the condition of patients under Enzyme Replacement Therapy (ERT), we evaluated the nutritional status of a relatively large sample of GD patients followed at Federico II University Hospital in Naples, Italy. (2) Methods: The study, having a cross-sectional design and involving 26 patients on ERT, included routine biochemical analyses, bioelectrical impedance analysis, indirect calorimetry, and administration of food frequency and physical activity questionnaires. The results in GD patients were compared with those from an appropriate control group. (3) Results: GD patients had normal biochemical parameters in 80% of cases, except for HDL-cholesterol, consumed a hyper-lipidic diet, and had a 60% prevalence of overweight/obesity. Body composition did not differ between patients and controls; however, measured REE was significantly lower than predicted and was reduced in comparison with the healthy controls. (4) Conclusions: This study provided novel elements to the present knowledge about REE and the nutritional status of GD patients under ERT. Its results warrant confirmation in even larger GD population samples and a more in-depth investigation of the long-term effects of treatment superimposed on the basic pathophysiological disease condition.

## 1. Introduction

The Gaucher disease (GD) (OMIM ID: 230800) is an autosomal recessive lysosomal storage disease caused by the reduced activity of the lysosomal enzyme β-glucocerebrosidase (GCase), leading to the accumulation of the glycosphingolipid substrate, glucosylceramide (GL-1), and its derivative glucosylsphingosine (Lyso-GL1), particularly in macrophages. Very rarely, GD could be caused by saposin C deficiency. The incidence of GD ranges between 1/40,000 and 1/60,000 births in the general population, but in the Ashkenazi Jewish population, it reaches 1/800 births [1,2]. The accumulation of glycosphingolipid substrate is prevalent in the monocyte-macrophage cells of the liver, spleen, and bone marrow [1,2]. Three phenotypes of GD are currently recognized. Type 1 GD (GD1), the most frequent, is characterized by variable clinical manifestations, ranging from asymptomatic to life-threatening forms, but rarely with neurological impairment. The most common clinical signs are splenomegaly, observed in more than 90% of patients, hepatomegaly, thrombocytopenia, anemia, fatigue, and bone involvement. Type 2 GD is the most severe form showing different phenotypes such as prominent visceral involvement, hydrops fetalis, congenital ichthyosis, thrombocytopenia, growth retardation, strabismus, dysphagia, and severe neurological involvement. This form is very rare and most often causes death in the first 2 years of life. Type 3 GD affects children, adolescents, and young adults with a less severe neurological involvement characterized by supranuclear gaze palsies, ataxia, a central defect of auditory processing, myoclonus, and occasionally seizures.

Before the introduction of enzymatic replacement therapy (ERT), GD was characterized from the metabolic point of view by a systemic inflammatory state, higher resting energy expenditure (REE) [3], and risk of malnutrition, especially in childhood. Conversely, ERT-treated adult GD1 patients seem to be at increased risk of overweight and obesity [4].

ERT, introduced in the early 1990s, modified the natural history of the disease and significantly improved the life expectation and the quality of life of GD patients, reducing visceral, hematological, and bone complications. Not surprisingly, this has led to an increased risk for these patients to develop the usual comorbidities linked to unhealthy lifestyles and food habits, namely overweight and obesity, impaired glucose metabolism [5,6], and insulin resistance [7].

The few studies that analyzed the energy metabolism and the REE of GD1 patients suggested a hypermetabolic state [3,5,6,8] in naïve patients, possibly due to the inflammatory condition, that ERT partially reduced [6,8]; however, definitive conclusions in this regard are difficult to be drawn because of the heterogeneity of the examined populations and the small or very small samples investigated in the available studies. Therefore, the aim of the present study was to evaluate the nutritional status and, in particular, the REE of the GD1 patients followed at the Reference Centre for diagnosis and treatment of Gaucher Disease in the adult population at the Department of Clinical Medicine and Surgery of “Federico II” University Hospital in Naples, Italy.

## 2. Materials and Methods

### 2.1. Inclusion Criteria

Inclusion criteria were: (a) diagnosis of Gaucher disease, confirmed by glucocerebrosidase activity assay and glucosylceramidase beta 1 (GBA1) gene analysis (reported in Appendix A); (b) classification type: GD type 1; (c) age > 18 years; (d) having undergone a follow-up visit at the Reference Centre in 2021; (e) having signed the consent form to participate in the study.

### 2.2. Study Design

This cross-sectional study included 26 GD1 patients seen between February and June 2021. The study was conducted according to the Declaration of Helsinki and approved by the Federico II Ethical Committee (Protocol’s number: 209/20). The patients were seen early in the morning, and all measurements were conducted after a fasting period of 12h according to standardized conditions, i.e., abstaining from alcohol and vigorous physical activity (PA) for 24 h prior to the assessment.

The investigations included: (a) the collection of demographic characteristics, (b) the routine biochemical analyses, (c) the analysis of body composition by bioelectrical impedance analysis (BIA), (d) the estimation of REE by indirect calorimetry, (e) the evaluation of food habits and nutritional intakes by the administration of a Food Frequency Questionnaire (FFQ) validated for the Italian population [9] and the subsequent calculation of nutrient intakes, (f) the evaluation of the habitual physical activity level by the administration of the short form of the International Physical Activity Questionnaire [10], and (g) the calculation of the duration of ERT and drug treatment.

### 2.3. Control Group

The patients’ data were compared with those obtained in a sample of 78 healthy individuals (1 case: 3 controls) matched by age, sex, and BMI. This sample was randomly selected from our database of healthy individuals who had undergone a body composition analysis and indirect calorimetry during the same period (February–June) of the last 3 years (Caucasian adults, aged between 18 and 70 years).

### 2.4. Anthropometry

Body weight and height were measured to the nearest 0.1 kg and 0.5 cm, respectively, with subjects wearing light clothes and no shoes, using a platform beam scale with a built-in stadiometer (Seca 709; Seca, Hamburg Germany). BMI was calculated as body weight expressed in kilograms divided by squared height reported in meters.

### 2.5. Bioelectrical Impedance Analysis

Bioelectrical Impedance Analysis was performed at 50 kHz (Human Im Plus II, DS Medica, Milan, Italy) at a constant room temperature of 22–25 °C. Measurements were carried out on the non-dominant side of the body, in the postabsorptive state, after voiding and with the subject in the supine position for 20 min. The BIA variables considered were resistance (R) and reactance (Xc). The Phase angle (PhA) was calculated as arc tangent Xc/R × 180°/π and expressed in degrees. A bioimpedance index (BI-index) was defined as squared height, reported in centimeters, divided by R (cm^2^/W).

Finally, fat-free mass (FFM) and fat mass (FM) were estimated by using the predictive equations developed by Sun et al. [11]

### 2.6. Resting Energy Expenditure Assessment

To measure REE, indirect calorimetry (IC) with a canopy system (Vmax 29 and Vmax Encore, Sensor Medics, Anaheim, CA, USA) was used [12]. After an overnight fast (at least 12 h) and 10 min of adaptation time, the patients laid down on a bed in a quiet room (at a constant temperature of 22–25 °C), and oxygen consumption and carbon dioxide production were measured for 30 min. The first 10 min were discarded for data analysis. All patients were evaluated between 4 and 11 days after the last enzyme replacement treatment to avoid any immediate potential treatment effects on REE. Energy expenditure was calculated using the abbreviated Weir’s formula, neglecting protein oxidation [13].

REE was also estimated using 2 predictive equations: the standard and universally used Harris–Benedict equation for normal weight subjects [14] and a validated equation previously developed for the adult population of Southern Italy [15].

### 2.7. Assessment of the Habitual Physical Activity Level

The International Physical Activity Questionnaire Short-Form (IPAQ-SF) is a validated, widely used self-report questionnaire to assess the habitual PA level [10]. It consists of seven questions aiming to capture the average daily time spent sitting, walking, and involving in moderate and intense PA over the last seven days. The IPAQ-SF was self-administered providing the patients with standardised instructions [16]. Assistance was provided by the researchers (PII and ES) to read the questions if required. The final score was used to categorize PA level into 3 groups: (1) inactive, (2) sufficiently active, and (3) active or very active.

### 2.8. Evaluation of Food Habits

Quality and quantity of food intake were estimated by the administration of a validated food frequency questionnaire (FFQ) for the Italian population [9]. Patients were asked by trained nutritionists (PII and ES) about their consumption of 36 different food items based on commonly eaten foods and portion sizes of the food list. Subjects were also asked to indicate how often, on average, they had consumed foods and drinks included in the FFQ, ranging from ‘‘never’’ or ‘‘less than once a week’’ to ‘‘seven times per week’’.

The food items were categorized as drinks, milk and dairy products, meat, fish, eggs, cereals, vegetables, legumes, fruit, fatty dressings, and others (sweets, fried foods, and fast food). For coffee, alcoholic and soft drinks, the amount in terms of cups/glasses was also specified.

From the record of food items, the daily intake of macronutrients was calculated using the WinFood software (version 3.9, Medimatica Srl, Colonella (TE), Italy) which refers to the European Institute of Oncology and INRAN Food Composition Tables. Carbohydrates, protein, and fat intakes were classified as a percentage of total daily energy intake.

### 2.9. Biochemical Parameters

Venous blood samples were taken for the measurement of several potential biomarkers of nutritional status, i.e., albumin (g/dL), haemoglobin (g/dL), total cholesterol (mg/dL), LDL-cholesterol (mg/dL), HDL-cholesterol (mg/dL), total lymphocytes (10^3^/mL), total protein (g/dL), aspartate and alanine aminotransferase (U/L), platelet count (10^3^/UL), glucose (mg/dL), insulin (µU/mL), vitamin B12 (pg/mL) and folate (ng/mL). Chitotriosidase (CT), a surrogate plasma marker for Gaucher disease activity and responsiveness to the ERT, was measured in all patients. The enzyme is released by storage cells and is elevated in serum of Gaucher patients. Plasma CT level is measured with the substrate 4-methylumbelliferyl (4MU)-chitotriose [17]. Glucosylsphingosine (lyso-Gb1) was measured by tandem mass spectrometry from Dried Blood Spot. 

All determinations were performed at the central laboratory of Federico II University Hospital using standardized techniques.

### 2.10. Blood Pressure Measurement

Systolic (SBP) and diastolic blood pressure (DBP) were measured with automatic validated devices (OMRON 705IT) after having the participant sit for at least 10 min. With the patient in the sitting position, 3 blood pressure measurements were recorded, 1–2 min apart, and additional measurements were collected only if the first two readings differed by >10 mmHg. BP was recorded as the average of the last two BP readings. 

### 2.11. Spleen and Liver Volumes

Spleen and liver volumes were measured by Nuclear Magnetic Resonance (NMR) or by abdominal ultrasound examination in 7 patients who refused to undergo NMR. In patients who underwent NMR, multiples of normal volumes for spleen (0.2% of body weight) and liver (2.5% of body weight) were calculated [18,19].

### 2.12. Statistical Analysis

The statistical analysis was carried out using the SPSS for Windows, version 27 (SPSS inc, Chicago, IL, USA). The descriptive statistics covered both the whole study population and stratification by sex. Results were expressed as mean and standard deviation (SD). Differences between groups were assessed with unpaired *t* tests. Two-sided *p* values less than 0.05 were considered statistically significant.

## 3. Results

### 3.1. Study Participants

Twenty-six GD1 patients (15 women and 11 men) were included in the analysis. All patients (except for one) received ERT by venous infusion every 2 weeks (Imiglucerase or Velaglucerase). The untreated patient showed no significant clinical signs of GD or altered chitotriosidase values. The mean duration of therapy was 16.4 years, from a minimum of 3 to a maximum of 27 years. The mean age was 50.6 ± 14.8 years, and the mean BMI was 27.6 ± 4.3, with no statistically significant differences between men and women. Table 1 shows the general and anthropometric characteristics of the study sample, whereas the biochemical values can be found in Appendix A.

Eighty percent of the patients featured normal biochemical parameters, except for HDL-cholesterol, which was low (<45 mg/dL) in 48.1% of the sample.

Eighteen patients out of the nineteen who underwent NMR had a normal hepatic volume (mean volume: 1455.9 ± 247.1 mL), while three out of seven who underwent abdominal echography had hepatomegaly (mild in two cases and moderate in one case). By contrast, only one of the patients who underwent NMR had a normal splenic volume, with a mean volume of 395.6 ± 185.1 mL (on average 2.17-fold larger than normal), and one patient who underwent abdominal echography had moderate splenomegaly. Finally, three patients had undergone splenectomy.

### 3.2. Body Composition and Resting Energy Expenditure

Table 2 shows the body composition and the REE (measured and estimated) in the GD1 population. All values were statistically different between men and women. Measured REE was lower than expected using both the Harris–Benedict and the Marra equations, with a significant difference of 116.6 ± 152.2 Kcal/die (−7.6%) between the measured REE and the value predicted by the Harris–Benedict equation (*p* < 0.0001).

Likewise, comparing the REE of GD1 patients and the REE of the healthy controls (33 men and 45 women) matched by age and BMI, both male and female GD1 patients were found to be hypometabolic (Table 3 and Table 4). In male patients (Table 3), REE was 9.3% lower than in healthy male controls, whereas in female GD1 patients (Table 4), REE was 11.8% lower compared to female controls. A linear correlation was detected between REE (measured and estimated) and FFM in GD patients and controls (Figure 1).

A direct correlation was also found between REE and hepatic volume (r = 0.577; *p* < 0.005) whereas no correlations were found between REE and: age (r = −0.206; *p* = 0.3), BMI (r = 0.292; *p* = 0.1), IPAQ score (r = −0.016; *p* = 0.9), splenic volume (r = 0.220; *p* = 0.3), duration of treatment (r = −0.186; *p* = 0.4), and treatment dosage (r = −0.271; *p* = 0.2). Moreover, no correlations were found between fat mass (%) and both duration of treatment (r = 0.217; *p* = 0.3) and ERT dosage (r = 0.154; *p* = 0.5). Finally, no correlations were found between fat-free mass (kg) and both duration of treatment (r = −0.226; *p*= 0.3) and ERT dosage (r = 0.173; *p* = 0.4).

### 3.3. Food Habits and Physical Activity Level of GD1 Patients

The analysis of the patients’ energy and macronutrient intake showed a tendency toward an unbalanced diet because of a hyper consumption of lipids, based on the Dietary Reference intakes for energy and macronutrients for the Italian population [20] (Figure 2).

Based on the IPAQ-SF, 42% of the patient population was inactive, 27% sufficiently active, and 31% active or very active (Table 5).

## 4. Discussion

This is, to the best of our knowledge, the first study on the nutritional status and energy metabolism in Italian GD1 patients and the study with the largest adult sample size of patients on Enzyme Replacement Treatment. The main findings of the study are that:GD1 patients on ERT had normal biochemical parameters in 80% of cases, except for HDL-cholesterol, which was lower than the normal cut-off in almost 50% of the population;A total of 60% of GD1 patients were overweight or obese;GD1 patients consumed a hyper-lipidic diet, and almost half of the population was inactive;Body composition did not differ between patients and healthy controls;REE measured by indirect calorimetry was significantly lower than predicted by the Harris–Benedict equation. This was confirmed using a specific equation for the REE estimation of Southern Italy residents;Measured REE was also reduced if compared to healthy controls;REE was directly related to hepatic volume in GD1 patients.

The observation of normal biochemical parameters may not be surprising, being a result of the effect of ERT, as observed in previous studies [4], except for HDL-cholesterol, which was seen persistently low after 3 years of treatment also by Zimmermann et al. [21]. Although in those studies, HDL-cholesterol had increased by 38% during ERT, the low HDL-cholesterol value was considered a potential risk factor for atherosclerotic changes. This condition has also been described in another rare metabolic disease [22].

The prevalence of overweight/obesity in our sample was very similar to that described by Kaluzna et al. [4] (56%) in a systematic review of the endocrine and metabolic disorders of GD1 patients. This prevalence is similar to the one seen in the general population [7].

According to the FFQ GD1, patients’ diet is characterized by a high percentage (>40%) of lipids. This reflects the westernization of the global diet, and it is coherent with the results of a survey conducted on a representative sample of the Italian population of 8462 individuals in 2008–2012 through the EPIC questionnaire [23]. The correlation between hepatic volume and REE, although the liver was of normal dimensions in almost all of our patients, testifies to the major metabolic involvement of the liver in the determination of the basal metabolic rate (BMR), which, according to Koff et al. [24], it explains as much as 21%.

The most surprising result of our study was the hypometabolic condition of our treated GD1 patient population. This finding is in contrast with the results of the other authors who measured REE by indirect calorimetry in GD1 patients [3,5,6,8] and found, in general, an excess REE.

Barton et al. [3] measured REE in GD1 patients before the introduction of ERT as a therapeutic option. In their study of 25 Israeli patients (14 adults and 11 children/adolescents), of which 18 were Ashkenazy, REE exceeded by 44% the value estimated by the Harris–Benedict equation. Similarly to our study nevertheless, REE was directly related to liver volume (ρ = 0.78; *p* < 0.05). In that study, excess REE decreased after splenectomy (five patients), a finding not confirmed by our study, in which three splenoctemized patients had an REE about 100 Kcalories higher than the rest of the sample. REE was also measured in two patients with other lysosomal storage diseases and found to be normal.

Corssmit et al. [5] in 1995 compared REE measurements in seven adult Danish non-treated GD1 patients and seven healthy controls matched by age, weight, and lean body mass. This study showed an REE 24% higher in patients than in controls, but no comparison was made between measured and estimated REE.

Hollak et al. [6] in 1997 analyzed the same population as Corssmit et al. They measured REE in 12 Danish patients before and after 6 months of ERT. All patients presented with organomegaly, which was reduced by 10% for the liver and 20% for the spleen upon treatment. REE before ERT was 29% higher than estimated by the Harris–Benedict equation; after 6 months, it decreased by 9%, while weight and fat mass significantly increased (*p* < 0.02). These authors did not find any correlation between REE decrease and changes in organ volumes

Doneda et al. [8] in 2011 analyzed 14 GD1 Brazilian patients (10 on ERT for a mean period of 5 years and 4 on no treatment). Measured REE was 27.1% higher in GD1 patients on ERT compared to controls (*p* = 0.007), while it was 16.2% higher in GD1 in the four untreated patients (*p* = 0.32). No significant difference in the REE of treated and untreated patients was observed. Moreover, two of the latter agreed to repeat indirect calorimetry after 6 and 8 months of ERT. In both patients, a further increase was observed in REE. Additionally, in this study, no correlation was found between REE and hepatomegaly, while a correlation was found between REE and BMI.

Although all previous studies seem to agree on a hypermetabolic condition of GD1 patients, both before and during ERT, they showed contrasting results with regard to the relationships of REE with organomegaly, BMI, and ERT. The large heterogeneity of the populations of these studies (different ethnic groups, different ages, different mean duration of therapy, presence or absence of ERT, different type of ERT when present, different prevalence of organomegaly) together with the very small sample sizes makes an explanation very difficult. A hypometabolic condition, in general, has been shown to be caused by different factors such as aging, which causes a loss of free fat mass, or clinical conditions such as malnutrition, cachexia, or hypothyroidism, or the use of drugs (i.e., antianxiety and antipsychotic agents). Neither of these factors, however, apply to our patient population. Indeed, a possible explanation of the hypometabolic state of the GD1 patients may be their physical inactivity, which, together with their bones and joints pain, may have caused a partial reduction of REE. In order to validate this hypothesis, it could be useful to monitor the patients’ daily physical activity by a recently developed wearable device using a mobile phone application. This recent technology has been tested in a pilot study in patients with Gaucher disease [25].

Our work has several strengths. In addition to being, to our knowledge, the first assessment of the nutritional habits and nutritional status in Italian GD1 patients, it is also the largest analysis of the REE of adult GD1 patients worldwide. The measurement of REE was based on indirect calorimetry performed under accurately standardized conditions. The REE of GD1 patients was compared with that of a substantial number of healthy individuals, accurately matched by age, sex, and BMI, and was also compared with the one estimated by the Harris–Benedict equation. In addition, an additional equation specifically developed and validated for Southern Italy residents further supports our results.

Our work also has some limitations. The major one is that we did not have the composition of the diet of the healthy controls and their PA level. Moreover, the sample, although much larger than that of other studies on the REE of GD1 patients, was nevertheless still not large enough to draw definitive conclusions.

## 5. Conclusions and Perspectives

In our opinion, this study’s results are a useful addition to the present knowledge about the resting energy expenditure and the nutritional status of GD1 patients. At the same time, our study clearly indicates the need for further collaborative research allowing the achievement of much larger sample sizes and the comparison of findings obtained in different populations and at different times relative to the initiation of ERT. The problem represented by the scarcity of patients due to the rare nature of the disease could be overcome by the development of multicentre national studies, a process recently undertaken by the Italian Society of Human Nutrition in collaboration with other scientific societies with the aim to extend the study of nutritional status and energy metabolism to a larger number of GD1 patients recruited in different Italian regions.

## Figures and Tables

**Figure 1 nutrients-14-03180-f001:**
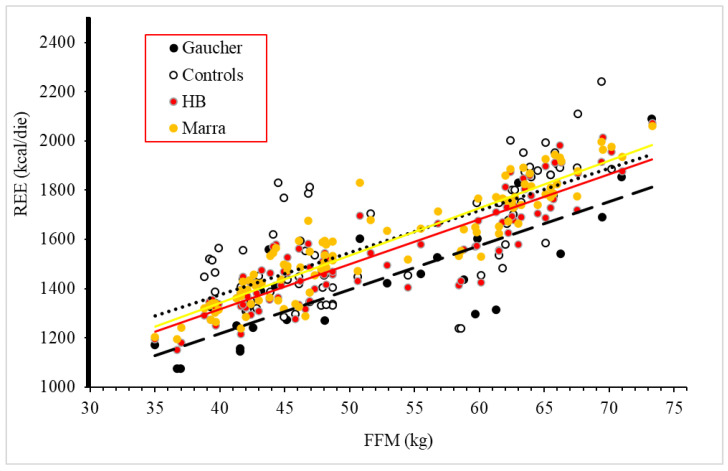
Relationship between fat-free mass (FFM) and resting energy expenditure (REE) in GD1 patients and healthy controls. Gaucher: Gaucher patients. Controls: healthy controls. HB: estimation by Harris-Benedict equation. Marra: estimation by Marra equation.

**Figure 2 nutrients-14-03180-f002:**
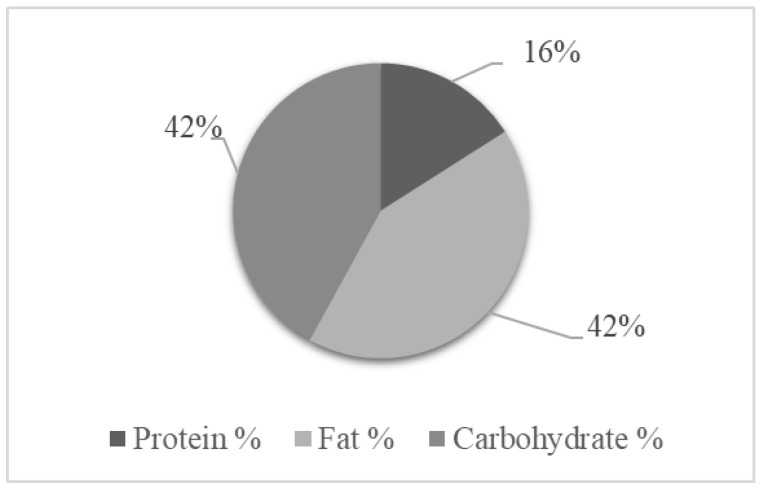
Percent energy from macronutrients in the study population of patients with type I Gaucher disease (total energy intake 2091 kcal/day).

**Table 1 nutrients-14-03180-t001:** General and anthropometric characteristics of the study population.

	Whole Population (*n* = 26)	Men (*n* = 11)	Women (*n* = 15)
	Mean ± SD	Mean ± SD	Mean ± SD
Age (years)	50.6 ± 14.8	52.6 ± 9.2	49.1 ± 18.1
DOT (months)	189 ± 96	160 ± 102 *	209 ± 89
Body weight (kg)	74.9 ± 14.3	83.8 ± 11.0 *	68.4 ± 13.1
Height (cm)	165 ± 10	174 ± 6 *	158 ± 7
BMI (kg/m^2^)	27.6 ± 4.3	27.8 ± 3.5	27.5 ± 4.9
Waist circumference (cm)	100 ± 10	100 ± 8	101 ± 12
Hip circumference (cm)	89.1 ± 10.7	97.1 ± 6.9	83.3 ± 9.3
Systolic blood pressure (mmHg)	128 ± 13	134 ± 11	125 ± 13
Diastolic blood pressure (mmHg)	80 ± 9	84 ± 9	76 ± 7

* *p* < 0.05 vs. women. DOT—duration of treatment; BMI—body mass index; SD—standard deviation.

**Table 2 nutrients-14-03180-t002:** Body composition and resting energy expenditure (measured and estimated) of the study population.

	Whole Population (*n* = 26)	Men (*n* = 11)	Women (*n* = 15)
	Mean ± SD	Mean ± SD	Mean ± SD
FFM (kg)	51.5 ± 11.5	63.2 ± 6.0 *	42.9 ± 5.0
FM (kg)	23.5 ± 8.2	20.6 ± 6.2	25.5 ± 9.1
FM (%)	31.3 ± 8.3	24.2 ± 4.7 *	36.4 ± 6.4
Phase angle (degrees)	6.14 ± 0.86	6.67 ± 0.67 *	5.75 ± 0.79
REE (kcal/day)	1422 ± 251	1603 ± 244 *	1290 ± 161
REE-HB (kcal/day)	1539 ± 250	1746 ± 189 *	1386 ± 165
REE-Marra (kcal/day)	1574 ± 252	1793 ± 150 *	1413 ± 179

* *p* < 0.05 vs. women, FFM—free-fat mass; FM—fat mass; REE—resting energy expenditure measured by indirect calorimetry; REE-HB—resting energy expenditure estimated by Harris–Benedict equation; REE-Marra—resting energy expenditure estimated by Marra equation; SD—standard deviation.

**Table 3 nutrients-14-03180-t003:** Body composition and resting energy expenditure (measured and estimated) of male GD1 patients (*n* = 11) compared to male healthy controls (*n* = 33).

	Male Patients *n* = 11	Male Controls *n* = 33	*p*
Age (years)	52.6 ± 9.2	52.4 ± 8.1	0.942
Body weight (kg)	83.8 ± 11.0	82.6 ± 9.1	0.708
Height (cm)	174 ±6	172 ± 6	0.488
BMI (kg/m^2^)	27.8 ±3.6	27.8 ± 2.5	0.984
FFM (kg)	63.2 ± 6.0	63.5 ± 3.2	0.837
FM (kg)	20.6± 6.2	19.1 ± 6.9	0.513
FM (%)	24.2 ± 4.7	22.5 ± 6.2	0.410
Phase Angle (degrees)	6.67 ± 0.67	6.60± 0.43	0.709
REE (kcal/day)	1603 ± 244	1767 ± 226	**0.047**
REE-HB (kcal/day)	1746 ± 189	1724 ± 159	0.701
**Δ REE vs. HB (Kcal/day)**	**−143 ± 192**	**43 ± 129**	**0.001**
REE-Marra (kcal/day)	1793 ± 150	1773 ± 129	0.668
**Δ REE vs. Marra (Kcal/die)**	**−190 ± 181**	**6 ± 143**	**0.001**

BMI—body mass index; FFM—free-fat mass; FM—fat mass; REE—resting energy expenditure measured by indirect calorimetry: REE-HB—resting energy expenditure estimated by Harris–Benedict equation; REE-Marra—resting energy expenditure estimated by Marra equation. In bold the difference between measured REE and estimated ones.

**Table 4 nutrients-14-03180-t004:** Body composition and resting energy expenditure (measured and estimated) of female GD1 patients (*n* = 15) compared to female healthy controls (*n* = 45).

	Female Patients *n* = 15	Female Controls *n* = 45	*p*
Age (years)	49.1 ± 18.1	47.9 ± 9.5	0.742
Body weight (kg)	68.4 ± 13.1	70.0 ± 9.6	0.608
Height (cm)	158 ± 7	159 ± 6	0.620
BMI (kg/m^2^)	27.5 ± 4.9	28.0 ± 5.1	0.746
FFM (kg)	42.9 ± 5.0	44.6 ± 3.4	0.135
FM (kg)	25.5 ± 9.1	25.4 ± 7.9	0.964
FM (%)	36.4 ± 6.3	35.6 ± 6.8	0.664
Phase Angle (degrees)	5.75 ± 0.79	5.74 ± 0.77	0.981
REE (kcal/die)	1290 ± 161	1462 ± 142	**0.000**
REE-HB (kcal/day)	1386 ± 165	1410 ± 86	0.485
**Δ REE vs. HB (Kcal/day)**	**−97 ± 119**	**52 ± 150**	**0.001**
REE-Marra (kcal/day)	1413 ± 179	1439 ± 109	0.513
**Δ REE vs. Marra (Kcal/die)**	**−123 ± 126**	**23 ± 160**	**0.002**

BMI—body mass index; FFM—free-fat mass; FM—fat mass; REE—resting energy expenditure measured by indirect calorimetry: REE-HB—resting energy expenditure estimated by Harris–Benedict equation; REE-Marra—resting energy expenditure estimated by Marra equation. In bold the difference between measured REE and estimated ones.

**Table 5 nutrients-14-03180-t005:** Classification of the study population by degree of physical activity based on the International Physical Activity Questionnaire Short Form (IPAQ-SF).

	Whole Population *n* = 26	Men *n* = 11	Women *n* = 15
Inactive	42.3% (*n* = 11)	36.4 (*n* = 4)	46.7% (*n* = 7)
Sufficiently active	26.9% (*n* = 7)	36.4 (*n* = 4)	20.0% (*n* = 3)
Active or very active	30.8% (*n* = 8)	27.3 (*n* = 3)	31.3% (*n* = 5)

*p* = 0.6.

## Data Availability

The data presented in this study are available on request from the corresponding author. The data are not publicly available due to privacy reasons.

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
