# Peer review of "Evaluation of the Nutritional Status of Gaucher Disease Type I Patients under Enzyme Replacement Treatment"

_nutrients, 2022, doi:10.3390/nu14153180_

Round 1
Reviewer 1 Report
In this article Authors investigated the nutritional status and in particular the resting Energy Expenditure (REE) of a group of GD patients under Enzyme Replacement Therapy (ERT) and compared them woth a group of healthy controls.
This is an interesting study that provides novel insights on the nutritional status and energetic metabolism in Italian GD1 patients on Enzyme Replacement Treatment, a superimposed treatment on the basic pathophysiological disease condition.
I write down here my observations.
1) The paper is generally well-structured and well-written. Please check typos errors throughout the text
2) The manuscript includes a detailed analysis of the current knowledge in this field with quite extensive literature survey.
3) Methods and Statistical analysis are adequate.
4) Ethics board approval, disclosure of funding and conflicts of interest are reported in the manuscript.
5) Comprehensive and concise conclusions.
6) The main limitation of the study is the small sample size, however authors reported this issue in the text.
Author Response
Dear Reviewer thank you very much for your comments. We are very pleased that you appreciated so much our work.
Thank you for having noticed typos, which have been corrected.
Reviewer 2 Report
The manuscript needs extensive revision.
Inclusion criteria -> Biochemical diagnosis -> Do You mean reduced GCase activity only ? What about molecular confirmation ? And what about lyso-Gb1 used as a biomarker in GD ?
3 out of 7 who underwent abdominal echography had a hepatomegaly -> This sub-group should be more extensively described regarding the fact that ERT normalizes the liver volume.
3 patients had undergone splenectomy -> before ERT introduction ?
I think that it will be of a great value to add some parameters, especially features of metabolic syndrome.
Body composition and resting energy expenditure -> it should be analyzed in terms of time and dosage of ERT.
Finally, the paper lacks some statistical analyses.
Author Response
Dear Reviewer, thank you for your comments.
The manuscript needs extensive revision. We have done a manuscript revision, following your suggestions.
Inclusion criteria -> Biochemical diagnosis -> Do You mean reduced GCase activity only ? What about molecular confirmation ? And what about lyso-Gb1 used as a biomarker in GD ? Thank you for this question that allows us to better clarify the diagnostic algorithm. For all patients, when Gaucher disease was suspected on the basis of the clinical picture, the enzymatic activity of GBA was measured and subsequently confirmatory molecular analysis was performed. We added lines 70-72 and lines 156-157.
3 out of 7 who underwent abdominal echography had a hepatomegaly -> This sub-group should be more extensively described regarding the fact that ERT normalizes the liver volume. Thank you for this question. Regarding the sub-group with hepatomegaly, 2 patients have only recently started ERT (3 years ago), therefore ERT has not completely normalized hepatic volume yet. In the third case the patient presented a very severe condition and hepatomegaly at the beginning of ERT, more than 20 years ago. So, the hepatomegaly has been mostly reduced during ERT, but not completely normalized.
3 patients had undergone splenectomy -> before ERT introduction ? Yes, splenectomy was performed in all the 3 patients before the diagnosis of GD. Moreover, for one patient the histological examination led to diagnostic suspicion of GD diagnosis. Finally, all splenectomies were carried out before ERT availability.
I think that it will be of a great value to add some parameters, especially features of metabolic syndrome. We added the parameter of blood pressure in table 1 and lines 162-168, while other biochemical analysis which can be used for the metabolic syndrome diagnosis are shown in table S1.
Body composition and resting energy expenditure -> it should be analyzed in terms of time and dosage of ERT. Finally, the paper lacks some statistical analyses. As suggested, we have added some statistical analyses (lines 235-238). In particular, we have analyzed the correlation between Resting energy expenditure (adjusted for fat free mass) and both years and dosage of ERT, and the correlation between fat free mass (% and kg) and fat mass with dosage and years of ERT. No correlation was found in any case. Here it follows schematically the correlation results:
|
Correlation |
r |
P |
|
REE -- ERT dosage |
-0.217 |
0.190 |
|
REE-- ERT years |
0.186 |
0.363 |
|
|
|
|
|
Fat mass (Kg) – ERT dosage |
0.319 |
0.120 |
|
Fat mass (%) – ERT dosage |
0.154 |
0.462 |
|
Fat mass (Kg) – ERT years |
0.106 |
0.605 |
|
Fat mass (%) – ERT years |
0.217 |
0.288 |
|
|
|
|
|
Fat free mass (Kg) – ERT dosage |
0.173 |
0.409 |
|
Fat free mass (Kg) – ERT years |
-0.226 |
0.266 |

Round 2
Reviewer 2 Report
The Authors responded to selected (by them) issues.
Still, there is no information about patients' genotype.
Authors added the following information: ,,Glucosylsphingosine (lyso-Gb1) was measured by tandem mass spectrometry from Dried Blood Spot''.
OK, what about correlations between Lyso-Gb1, its changes during treatment, and REE ?
Author Response
Dear Reviewer, thank you for your further comments that allows us to clarify some aspects of the paper
Still, there is no information about patients' genotype.
As previously specified, when Gaucher disease was suspected on the basis of the clinical picture, all patients underwent to an enzymatic activity of GBA measurement and subsequently confirmatory molecular analysis was performed. To better highlight the molecular analysis, a new table (S2) was added in the supplementary material.
Authors added the following information: ,,Glucosylsphingosine (lyso-Gb1) was measured by tandem mass spectrometry from Dried Blood Spot''. OK, what about correlations between Lyso-Gb1, its changes during treatment, and REE ?
Measurement of Lyso-GB1 levels is relatively new in our center, where instead chitotriosidase levels have been widely used as a biomarker of Gaucher disease. Therefore, also due to the increase in follow-up times related to the covid19 pandemic, we have the trend over time of Lyso-GB1 only for a small group of GD patients at moment. As expected, for these few patients lyso-GB1 levels decrease over time in association with ERT duration. However, for the purposes of this work, we analysed the correlation between Lyso-GB1 and REE only in the patients whose Lyso-GB1 levels were measured close to the nutritional data collection, as this is a cross-sectional analysis. Lyso-GB1 and REE are not related (r = 0.255; p = 0.280). Moreover, although unsurprisingly, a strong association was found between the levels of the Lyso-GB1 and chitotriosidase (r = 0.822; p <0.001) confirming their substantial overlap in the clinical setting.